Estimating relative risk of within-lake aquatic plant invasion using combined measures of recreational boater movement and habitat suitability

Wittmann Marion E. 1 2 3 mwittmann@gmail.com
Kendall Bruce E. 1
Jerde Christopher L. 3
Anderson Lars W.J. 4
1 Bren School of Environmental Science and Management, University of California , Santa Barbara, CA , USA
2 Department of Biological Sciences, University of Notre Dame , Notre Dame, IN , USA
3 Department of Biology, University of Nevada , Reno, NV , USA
4 Waterweed Solutions , Pt. Reyes, CA , USA
Larocque-Tobler Isabelle
Electronic publication date: 2015 Mar 19
Publication date: 2015
Volume: 3
Electronic Location ID: e845
Received 2014 Dec 19; Accepted 2015 Feb 28
Copyright: © 2015 Wittmann et al.
Copyright year: 2015
Copyright holder: Wittmann et al.
License: This is an open access article distributed under the terms of the Creative Commons Attribution License, which permits unrestricted use, distribution, reproduction and adaptation in any medium and for any purpose provided that it is properly attributed. For attribution, the original author(s), title, publication source (PeerJ) and either DOI or URL of the article must be cited.
License URL: https://creativecommons.org/licenses/by/4.0/

Keywords: Dispersal, Invasive species, Eurasian milfoil, Risk assessment, Wave action, Suitable habitat

Funding: University of California Center for Water Resources WR-1010 Institutional support from the Bren School of Environmental Science and Management, University of California, Santa Barbara This research was funded by the University of California Center for Water Resources Project Number WR-1010 and the Bren School of Environmental Science and Management at University of California, Santa Barbara. The funders had no role in study design, data collection and analysis, decision to publish, or preparation of the manuscript.

==============================
Effective monitoring, prevention and impact mitigation of nonindigenous aquatic species relies upon the ability to predict dispersal pathways and receiving habitats with the greatest risk of establishment. To examine mechanisms affecting species establishment within a large lake, we combined observations of recreational boater movements with empirical measurements of habitat suitability represented by nearshore wave energy to assess the relative risk of Eurasian watermilfoil (Myriophyllum spicatum) establishment. The model was evaluated using information from a 17 year (1995–2012) sequence of M. spicatum presence and absence monitoring. M. spicatum presence was not specifically correlated with recreational boater movements; however its establishment appears to be limited by wave action in Lake Tahoe. Of the sites in the “High” establishment risk category (n = 37), 54% had current or historical infestations, which included 8 of the 10 sites with the highest relative risk. Of the 11 sites in the “Medium” establishment risk category, 5 had current or historical M. spicatum populations. Most (76%) of the sites in the “Low” establishment risk category were observed in locations with higher wave action. Four sites that received zero boater visits from infested locations were occupied by M. spicatum. This suggests that the boater survey either represents incomplete coverage of boater movement, or other processes, such as the movement of propagules by surface currents or introductions from external sources are important to the establishment of this species. This study showed the combination of habitat specific and dispersal data in a relative risk framework can potentially reduce uncertainty in estimates of invasion risk.

Introduction

Predicting establishment for nonindigenous aquatic species (NAS) remains one of the greatest challenges for invasion ecologists yet is a key element for effective ecosystem monitoring and management. Assessing the risk of establishment requires an understanding of the number of individuals introduced to a particular area over time (i.e., propagule or colonization pressure) and characteristics of the receiving environment (Lockwood, Cassey & Blackburn, 2005). Where propagule pressure is high and habitat is suitable for a species to survive and reproduce, the risk of establishment and growth is substantial (VonHolle & Simberloff, 2005; Drake & Jerde, 2009).

Propagule pressure is difficult to measure directly for most aquatic species. Within aquatic ecosystems, boat movement as well as dispersal through natural currents contribute to the propagation and spread of aquatic species (Mosisch & Arthington, 1998; Beletsky et al., 2007; Clarke Murray, Pakhomov & Therriault, 2011). Both fragments and established populations of invasive seaweed (Caulerpa taxifolia) have been found in greater abundances in estuaries with high rates of recreational boating compared to areas with less recreational boating (West et al., 2009). Hull fouling associated with commercial or recreational activities is a well-known dispersal vector for both marine and freshwater introductions (Johnson & Carlton, 1996; Mineur, Johnson & Maggs, 2008; Clarke Murray, Pakhomov & Therriault, 2011). The exchange and discharge of ballast water has also been shown to increase secondary spread rates within the Great Lakes (Carlton, 1985; Sieracki, Bossenbroek & Faisal, 2013). Finally, advective movement via surface currents have also been found to increase the spread of both fish and invertebrate species within large lakes (Beletsky et al., 2007; Hoyer et al., 2014). While a lot of information exists regarding the spread of species within aquatic systems, the specific relationship between human-mediated and natural dispersal of species is largely unknown.

The focus of the current study was Eurasian watermilfoil (Myriophyllum spicatum), a freshwater macrophyte species native to Europe, Asia and Northern Africa and introduced to North America in the 1940’s (Couch & Nelson, 1985). In North America, M. spicatum impacts native species (Boylen, Eichler & Madsen, 1999) and has unwanted effects on ecosystem services (Eiswerth, Donaldson & Johnson, 2000; Halstead et al., 2003; CAST, 2014). Widespread dispersal occurred after its initial introduction in North America, as M. spicatum was a popular aquarium and trade species, and also planted into lakes and streams—spreading through water currents to connected waterways (Aiken et al., 1979; Madsen, Eichler & Boylen, 1988). Recreational boats have also been implicated as the main overland dispersal vector for freshwater aquatic plants, including M. spicatum (Johnstone, Coffey & Howard-Williams, 1985; Rothlisberger et al., 2010). However, other mechanisms of dispersion, such as endo- or ectozoochoric transport by birds have also been observed for aquatic plants (Figuerola & Green, 2002) M. spicatum is estimated to have established in this study system, Lake Tahoe, CA-NV [USA], in the 1970’s (Kim & Rejmankova, 2001; Anderson, 2003). By 2012 M. spicatum had spread to over 20 sites in water depths up to 5 m, covering approximately 0.34 km2 overall in the lake (Fig. 1).

Figure 1 Study site map figure.

Lake Tahoe, CA-NV. Circles indicate Eurasian watermilfoil (Myriophyllum spicatum) presence as of 2012. Crosses indicate wave action measurement sites.

The dispersion of M. spicatum within Lake Tahoe is a complex process with multiple interacting components. In Lake Tahoe, M. spicatum propagates primarily through vegetative fragments and not through seed germination (Walter, 2000). As most M. spicatum populations in Lake Tahoe are located within marinas or other nearshore protected zones, fragments are created when boat propellers cut the plant or when mechanical harvesting occurs (a non-chemical control activity in the lake). Fragmentation also occurs naturally in Lake Tahoe due to the plant’s phenology (e.g., autofragment production) (Barrat-Segretain & Bornette, 2000; Walter, 2000). An important factor of M. spicatum success as a colonizer is its ability to survive and produce roots up to  six weeks after fragmentation (Jerde et al., 2012; Mcalarnen et al., 2012).

Long distance dispersal within the lake then depends on transport mechanisms (e.g., entrainment on boats or boating equipment, advective transport through water currents, biologically based transport via birds or other species) between areas where M. spicatum is established and areas where it is not. This may include the movement of fragments across open waters or laterally within nearshore regions.

Once viable fragments reach a novel habitat, various environmental conditions such as temperature, sediment composition and energetics of surface waves may determine if new M. spicatum colonies will become established (Smith & Barko, 1990; Martin & Valentine, 2012). M. spicatum photosynthesizes and grows over a wide temperature range (15–35 °C) and can successfully overwinter in icy conditions (Smith & Barko, 1990). It grows best on fine textured inorganic sediments (Barko & Smart, 1986) but can be the dominant species over a wide range of sediment particle distributions (e.g., 15–100% sand) and sediment and/or water column nutrient concentrations (Smith & Barko, 1990; Madsen, 1999). The intensity of wave action and water movement are also important factors for M. spicatum establishment. Water flow may stimulate abundance at low to moderate velocities, but reduce growth at higher velocities (Schutten & Davy, 2000; Madsen et al., 2001; Martin & Valentine, 2012). Wave heights of 0.1–0.3 m have been shown to cause M. spicatum breakage, although not to the extent to impact viability of the plant (Stewart et al., 1997). As invasion success is dependent on multiple factors (e.g., transport, propagule pressure, habitat suitability), combining assessments of these factors, when possible, should improve estimates of risks for further spread and establishment.

Because it is difficult to observe an accurate relationship between propagule pressure and invasion risk when habitat suitability is different across sites, one approach is to include relative measures of individual survival and propagule pressure to develop a prediction framework (Herborg et al., 2007; Jerde & Lewis, 2007). For example, gravity models have used recreational boater movements to estimate relative abundance of human-transported NAS (Schneider, Ellis & Cummings, 1998; Bossenbroek, Kraft & Nekola, 2001; Muirhead & Macisaac, 2005), but have failed to incorporate the characteristics of the receiving habitat into predictions of establishment likelihood. Relative measures of species survival have been estimated using habitat matching models that compare species origins and putative destinations on a global scale (Drake & Bossenbroek, 2004; Herborg et al., 2007). We seek to combine measures of propagule pressure and habitat suitability within a lake, in order to establish a framework that can be used for managers tasked with minimizing the impact of invasion that is already ongoing.

This study assesses the relative risk of invasion spread within a single freshwater lake (Lake Tahoe) by examining two components of M. spicatum establishment: the physical properties of recipient habitats, and human-mediated propagule pressure via recreational boating trips between these habitats. We used direct measures of boater visitation frequency to approximate propagule pressure. Intensity of wave action at nearshore locations in Lake Tahoe was used to categorize relative risk into three categories (high, medium, low) and identify areas most vulnerable to recreational boat-mediated introduction of M. spicatum. We hypothesized that if wave height and propagule pressure are working in concert to determine establishment, then more sites in the high-risk category should be invaded than in the medium and low risk categories. Within a category, if propagule pressure is driving establishment, then sites with relatively more risk should be more likely to have been invaded.

Materials and Methods

Site description

Lake Tahoe is a large (surface area: 497 km2, max depth 501 m) oligotrophic lake located in the Sierra Nevada between California and Nevada USA at a subalpine elevation of 1,898 m. Measurements of water clarity in Lake Tahoe have shown average Secchi disk depths of 20 m (TERC, 2014) and light measurements of 1% light levels have been recorded to nearly 50 m (Rose et al., 2009). Since 1980, the volume-weighted annual average concentration of nitrate-nitrogen was 13–19 µg/L and that of total phosphorus was 1.5–4.0 µg/L (TERC, 2014). Annual average chlorophyll a in this same time period was 0.7–1.1 µg Chl a/L (Heyvaert et al., 2013). The Tahoe basin’s granitic geology, the lake’s large volume (150 km3) and small watershed (800 km2) explain the low nutrient concentrations and primary productivity rates (Goldman, 1988). In recent decades, Lake Tahoe has been subject to a number of environmental stressors such as development, atmospheric deposition, and other impacts related to human-use or climate-related change. Lake Tahoe is subject to intense recreational pressure, with over 3 million people visiting and over 20,000 trailered boats launched into the lake each year.

Previously, Lake Tahoe’s benthic zone was dominated by a number of Characeae, mosses, liverworts and filamentous algae species, which have been observed at depths up to 400 m (Frantz & Cordone, 1967; Caires et al., 2013). The native macrophytes Andean milfoil (M. quitense), Canadian waterweed (Elodea canadensis), coontail (Ceratophyllum demersum), Richardson’s pondweed (Potamogeton richardsonii) and leafy pondweed (Potamogeton foliosus) are found in Lake Tahoe. With the exception of one marina location (the “Tahoe Keys” which was built into a dredged wetland site), where C. demersum has been the most abundant macrophyte species at water depths <2 m, the nonnative M. spicatum has dominated the submersed aquatic plant community at water depths <5 m since the mid-1990’s. In the early 2000’s curlyleaf pondweed (P. crispus) established in the southern region of Lake Tahoe, and populations have rapidly expanded along the southern shore. Where P. crispus has established, it also dominates the native nearshore macrophyte community, and in some cases, has replaced M. spicatum populations, particularly in protected embayments, constructed marinas and disturbed (dredged) areas.

Distribution of Eurasian watermilfoil populations and recreational boater survey

Lake surveys to determine Eurasian watermilfoil distribution, 1995–2012

Whole-lake surveys for M. spicatum presence and absence were carried out annually in Lake Tahoe from 1995 to 1997 and in 2000, 2003, 2006 and 2012. A two- to three-person boat crew circumnavigated the nearshore zone, including marinas and other embayments, and visually inspected below the water surface for aquatic macrophytes from the vessel. If vegetation was spotted, a double-edged rake was thrown into the vegetation or divers snorkeled underwater to retrieve samples for species identification in the laboratory (Anderson & Spencer, 1996). In 2012, divers snorkeled or used SCUBA amongst vegetation to make in situ identification (K. Boyd, pers. comm., 2014).

Recreational boater survey

To determine the pathways of Lake Tahoe boaters, individuals (N = 778) were interviewed at public and private Lake Tahoe boat launches during the summer periods of 2005 and 2006 on 30 dates from July–September 2005 and June–September 2006. Of the 30 dates, 14 were weekdays, and 16 were weekends and/or holidays. On any given date, interviews were conducted for an 8–10 h period between 8 A.M. and 6:00 P.M. The interview consisted of ten questions and lasted approximately 5–10 min. Questions relevant to this study pertained to the boater’s launch origination and trips made between nearshore zones within the lake. The set of originations and destinations were defined by responses given by boaters, with as few as 1 and as many as 5 origination and destination combinations per boater collected. Each origination and destination combination was counted as one trip, and when the origination was from a site that contained M. spicatum, that trip constituted one potential propagule. This measurement of visitation to each boater destination site from a set of infested locations is referred to as B. A point biserial correlation coefficient was computed to assess the relationship between the presence of M. spicatum (including extirpated populations) and recreational boater visitation.

Habitat characterization

Wave action

To gauge the amount of energy or wave action in nearshore zones in Lake Tahoe, change in vertical pressure was measured using submersed depth pressure sensors (RBR DR-1050, accuracy ±0.05%) at 13 locations around the lake (Fig. 1). The sensor locations were distributed around all sides of the lake and were chosen to capture nearshore wave action caused by prevailing wind patterns (Schladow et al., 2012). Each sensor was placed at approximately the same depth (3 m) and set at a 1 s sampling interval for a period of 14 days from July through September 2006. Because there were only four sensors and a limited field period, measurements were taken continuously at the northern end of the lake (site CBI) with a single logger, and three other loggers were moved every 14-day sampling period. The continuous measurements taken at CBI were used to estimate significant wave heights (Hs, or the highest 1/3 of all waves measured) during the weeks for which a site did not have a logger present.

Change in surface water depth was calculated using the following pressure to wave height conversions: (1) pressure=p−Atmosphericpressure(dBar),

where p = pressure reading from the sensor (dBar), and atmospheric pressure was the calibration for high elevation conditions at Lake Tahoe (1,897 m). The conversion of pressure into depth was described by the following equation: (2) depthm=pressuregρ

where g is a gravitational constant (0.980665 m s−2) and ρ (1.0 g mL−3) is water density. To characterize the lake state in the various nearshore areas, significant wave heights (Hs), maximum wave heights (Hmax), and the root mean square wave heights (Hrms) were determined for all sites and represented the temporal variability over the entirety of the sampling period for each site (Dean & Dalrymple, 1991). For each of the locations identified by recreational boaters, wave height characterizations were assigned based on proximity to the nearest pressure sensor measurement.

Water column and sediment characteristics

Preliminary measurements at 23 sites around the lake of nearshore water column characteristics (chl a, dissolved oxygen, pH, temperature, and turbidity) and sediment nutrient and mineral concentrations (NH4, NO3, Ortho-P, TP, Ca, Mn, Fe) indicated no meaningful variation in these habitat features between sites; thus these variables were removed from the habitat assessment.

Estimating relative risk

We used relative measures of boater visitation from an infested site  (B), to assess invasion risk of M. spicatum within Lake Tahoe. After Jerde & Lewis (2007), we calculated the relative ratio (RR) of B for invasion of location X relative to B for invasion of location Y, where location Y was the location with the lowest (non-zero) B, for each site. Simply, RR was the proportion of boater visitation (BX) for a site, relative to the BY for the least visited site: (3) RR=BXBY.

As M. spicatum establishment has been shown to be limited by wave action (Schutten & Davy, 2000; Martin & Valentine, 2012), we further refined the relative risk evaluation based on empirical measurements of wave height as an indicator of habitat suitability. This serves to improve the ability to prioritize specific sites for surveillance by categorizing relative risk by high, medium and low establishment risk. Specifically, these establishment risk categories were divided into three groups according to their maximum wave height (Hmax) as measured during the June–August, 2006 period in Lake Tahoe: “High establishment risk” (<0.2 m) “Medium establishment risk” (0.2–0.3 m) or “Low establishment risk” (>0.3 m). Relative risk comparisons between sites in different establishment risk categories were not valid owing to the unknown relationship between specific values Hmax and establishment of any particular M. spicatum fragment.

We used a chi-squared test to determine whether there were differences between the frequencies of invasion (e.g., realized establishments of M. spicatum) for the “High,” “Medium,” and “Low” establishment risk categories. If there were no statistically significant differences between these categories, then we would proceed to test the explanatory power of the relative risk across all sites. However, if there were statistically significant differences between these categories, then logistic regression would be performed on each category (High, Medium, and Low) with number of boater visits as the explanatory variable. All analyses were carried out using R (v 2.13.0).

Results

Eurasian watermilfoil survey

In 1995, there were 13 nearshore sites in Lake Tahoe with M. spicatum presence. The number of sites with M. spicatum presence slowly increased, with 17 sites observed in 2000, 22 sites in 2003 and 26 sites in 2005. In 2011 there were 23 sites with M. spicatum presence, and in 2012 the number of occupied sites declined again, to 17 (Fig. 2), with a total coverage of approximately 0.35 km2, or 0.07% of Lake Tahoe’s area. The decrease in number of sites in 2011 and 2012 relative to previous years is a result of management (bottom barriers, dredging) and/or other causes of extirpation of localized populations (K Boyd, pers. comm., 2014).

Figure 2 Boater visitation and site infestation.

Invasion probability as a function of propagule pressure as represented by boater visitation from sites infested with M. spicatum in Lake Tahoe. Black circles indicate M. spicatum presence in 2012.

Recreational boater survey

There were a total of 65 sites named by the 778 interviewed recreational boaters as destinations within Lake Tahoe (Fig. 2). There were 1756 origination–destination trips and the most visited sites included Emerald Bay (a popular scenic destination; N = 273 trips) and Tahoe Keys (a destination with amenities e.g., gas, food, launch ramp; N = 214). Both of these sites have established M. spicatum populations; however the Tahoe Keys infestation is much greater, with dense stands reaching the water surface and directly adjacent to moored boats and in boat traffic lanes. There were four sites (23% of those with infestations) where boater visitation was 0, yet populations of M. spicatum have been present in those locations for a majority of the invasion record. Other popular sites visited were those with amenities (restaurants, gas stations) or are known as popular places to recreate. There were 769 origination-destination trips from locations with M. spicatum. There was no significant correlation between the presence of M. spicatum (including extirpated populations) and recreational boater visitation (rpb = 0.22, df = 63, p = 0.08).

Physical habitat and relative risk categorization

Similar to Lake Tahoe nearshore wave heights recorded during 2008–2009 summer and winter periods (which included one winter storm) (Schladow et al., 2012), wave heights measured in this study ranged from 0 to 0.5 m (Table 1). In general, the eastern shore of Lake Tahoe receives more wave action than the west shore of the lake (Schladow et al., 2012). Pressure sensor measurements also confirmed this to be true during the summer of 2006; the highest maximum wave heights recorded were on the east or northeast shore at CR, CBI, RHP, SH and ZPH (Table 1). Of 13 sites measured, five sites had an Hmax < 0.2, four sites were between 0.2 and 0.3, three were 0.3 or greater and one sensor malfunctioned during its deployment at location DLB and was not included. This breakdown was used to define the establishment risk categories (e.g., Hmax < 0.2 = “High,” 0.2 < Hmax < 0.3 = “Medium” and Hmax > 0.3 = “Low”).

Table 1 Wave height measurements.

Location and position of pressure sensors in Lake Tahoe to measure nearshore wave heights June–August, 2006.

ID	Location name	Lat	Long	Hs	H max	H rms	Risk	
BWM	Boatworks Marina	39.171	−120.137	0.006	0.027	0.003	High	
KBG	Garwoods	39.225	−120.083	0.004	0.031	0.002	High	
CRM	Camp Richardson	38.939	−120.039	0.019	0.113	0.011	High	
LFL	Lake Forest Launch	39.181	−120.120	0.013	0.128	0.008	High	
EPM	Elks Point	38.984	−119.957	0.020	0.181	0.012	High	
ZPH	Zephyr Cove	39.007	−119.950	0.027	0.208	0.017	Medium	
RHP	Round Hill Pines	38.990	−119.954	0.025	0.213	0.016	Medium	
RUB	Rubicon Bay	39.002	−120.102	0.018	0.218	0.010	Medium	
SPE	Sugar Pine/Ehrman	39.060	−120.113	0.034	0.253	0.021	Medium	
SH	Sand Harbor	39.201	−119.931	0.029	0.294	0.019	Low	
CBI	Crystal Bay/Incline	39.248	−119.989	0.029	0.377	0.019	Low	
CR	Cave Rock	39.042	−119.949	0.059	0.537	0.040	Low	
Notes.

Hs Significant Wave Height

Hmax Maximum Wave Height

Hrms Root mean square Wave Height, all represented in meters (m).

Risk Category of Eurasian watermilfoil risk of establishment based on Hmax

High <0.2, 0.2 < Medium <0. 3, and Low >0.3 m.

There was a significant association between establishment risk category and frequency of M. spicatum presence (χ2 = 8.66, df = 2, p = 0.013; Table 2). Of the 37 sites in the “High” establishment risk category, 54% have current or historical infestations of M. spicatum, including 8 of the 10 sites with the highest RR in this risk category. Of sites in the “High” establishment risk category, 35% had B = 0, indicating no visitation by boaters originating from sites with M. spicatum. Of the 11 sites in the “Medium” establishment risk category, 5 have either current or historical M. spicatum populations and 9 sites have B >0. Most of the sites in the “Low” establishment risk category are located on the east or northeast shore (e.g., the locations with higher wave action), and only two of them have current or historical M. spicatum populations. However, both of these populations are in protected areas (e.g., behind rock cribs or within a marina), and were not exposed to wave action of the other 15 sites. Thus, these locations may be considered as high energy (e.g., low establishment risk) environments that are overcome by protective barriers.

Table 2 Relative risk boater movement table.

Proportion of boater visits from sites with Eurasian milfoil (B; Total number of trips from infested locations, N = 769), and RR(B) or Relative Risk based on B for 65 nearshore sites in Lake Tahoe, USA. RR is relative to site differentiation of establishment risk categorization (High, Medium, Low) as determined by measurements of nearshore wave action.

Site	B	RR(B)	
High establishment risk (low wave action)	
Emerald Baya	0.22	169	
Lake Forest	0.16	124	
Tahoe Keysa	0.10	79	
Camp Richardsonb	0.08	58	
El Doradoa	0.04	27	
Sunnysideb	0.03	22	
Baldwin Beachb	0.02	15	
Tahoe Citya	0.01	10	
Garwoods Dock	0.01	7	
Hurricane Bay	0.00	2	
Kiva Beacha	0.01	5	
Kings Beach	0.01	4	
Ski Runa	0.01	4	
South Shorea	0.01	4	
Statelinea	0.01	4	
Ski Beach	0.00	2	
Timber Covea	0.00	2	
Carnelian Bay	0.00	1	
Cascade	0.00	1	
Larsons Beach	0.00	1	
Lester Beach	0.00	1	
Pope Beacha	0.00	1	
Tahoe Meadowsa	0.00	1	
Tahoe Tavernb	0.00	1	
Agate Bay	0.00	0	
Chinquapin	0.00	0	
Dollar Point	0.00	0	
Elks Point Beacha	0.00	0	
High Sierra Boat Co	0.00	0	
Lakelanda	0.00	0	
Nevada Beacha	0.00	0	
Skylandia Beach	0.00	0	
Tahoe Flatsb	0.00	0	
Tahoe Parkb	0.00	0	
Tahoe Pines	0.00	0	
Tahoe Vista	0.00	0	
Medium establishment risk (medium wave action)	
Meeks Baya	0.05	14	
Zephyr Coveb	0.04	11	
Rubicon Bay	0.03	9	
DL Bliss State Park	0.02	5	
Sugarpine Point	0.02	5	
Obexer’s Marinab	0.02	4	
Homewoodb	0.01	3	
Round Hill Pinesa	0.01	2	
Chambers Beach	0.00	1	
Marla Bay	0.00	0	
Tahoma	0.00	0	
Low establishment risk (high wave action)	
Sand Harbor	0.04	16	
Cave Rock	0.02	6	
Skunk Harbor	0.01	4	
Incline Village	0.01	2	
Hyatt	0.00	1	
Secret Harbor	0.00	2	
Dead Man’s Point	0.00	1	
Thunderbird Lodge	0.00	1	
Cal Neva	0	0	
Chimnea Beach	0	0	
Crystal Baya	0	0	
Glen Brook	0	0	
Hidden Beach	0	0	
Logan Shoalsb	0	0	
Lynbrook	0	0	
Snake Harbor	0	0	
Speedboat Beach	0	0	
Notes.

a Currently infested with Eurasian milfoil.

b Historical infestation of Eurasian milfoil.

There was only adequate power for logistic regression analyses (e.g., enough observations of M. spicatum presence) within the high establishment risk category, which indicated that RR was not a reasonable predictor of M. spicatum presence (z = 0.903, p = 0.367, df = 36). When risk categorizations are removed and RR was considered over all sites, it was also not a reasonable predictor of M. spicatum presence (z = 1.386, p = 0.166, df = 64).

Discussion

Similar to previous assessments of M. spicatum establishment at the landscape scale (Buchan & Padilla, 1999; Rothlisberger & Lodge, 2011), we have found that propagule pressure as represented by recreational boater visitation was not a significant explanatory factor of its presence within a lake. Further, characteristics of the receiving habitat, e.g., wave action, were found to be a limiting factor for M. spicatum establishment in Lake Tahoe. However, the extent to which boater movement is a singular useful predictor of M. spicatum in Lake Tahoe is not clear. While recreational boats may certainly play a role in the release and movement of M. spicatum, the plant’s distribution may be more dependent on alternative dispersal vectors (e.g., wind-driven surface currents, transport by birds), variation in temporal scales, or habitat limitations.

There were four sites (23% of those with infestations) where boater visitation was 0, yet populations of M. spicatum have been present in those locations for a majority of the invasion record. This indicates that either the boater survey data did not accurately represent visitation, or that another physical process such as the movement of propagules by surface currents is important. For example, boaters may not necessarily deliver a propagule to other nearshore sites, but rather boats may break M. spicatum stems with propellers and create fragments which are then liberated out into the lake, where they may be susceptible to advective transport by water currents to other nearshore zones (Anderson, 2003).

However, it is possible that recreational boating played an important role in the direct delivery of invasive plants through entrainment on boats or equipment (Rothlisberger et al., 2010) to popular and scenic sites such as Emerald Bay. Emerald bay is one of the few non-marina sites that contain M. spicatum in the lake. It is also the most highly visited area by boaters in Lake Tahoe; over 70% of surveyed boaters visited this location. The predominant south shore winds and water flows move eastward (Schladow et al., 2012), the opposite direction of Emerald Bay from most established M. spicatum populations (see Fig. 1). The abrupt appearance of the recently established non-native species, curlyleaf pondweed, at Emerald Bay suggests that some sort of long distance dispersal mechanism may be supporting the establishment of species in this area.

Wave action has been cited as an important factor for M. spicatum growth and establishment in Lake Tahoe and elsewhere (Walter, 2000; Madsen et al., 2001; Martin & Valentine, 2012). The energetics of highly wavy sites such as CR, ZPH and SH along the eastern shore combined with M. spicatum absence (with the exception of locations where there are protective rock cribs or marina structures) supports this notion. Despite the short duration of empirical data collection (e.g., 14 days per probe and a two month overall period) at each site and the interpolation of the measurements, these observations capture the range of multi-year wave heights (including summer and winter storms) both empirically measured Schladow et al., 2012) and simulated (Smith, 2001) in Lake Tahoe.

Temporal lags associated with the expansion of M. spicatum within Lake Tahoe may also be indicative of why some sites with high relative risk estimates do not have established M. spicatum populations. We propose that these lags may be attributed to the lake’s trophic status. First discovered in Lake Tahoe over 60 years ago, M. spicatum is currently established in only 17 locations around the 116 km lake perimeter, with an abundance of potentially suitable (e.g., sandy sediments and protected embayment) habitats remaining unoccupied. Oligotrophic systems, such as Lake Tahoe, often are characterized by low benthic taxon richness (Declerck et al., 2005), which may make these communities less resistant than more diverse communities to species invasions (Stachowicz et al., 2002). Properties of oligotrophic systems that contribute to low taxon richness, such as low nutrient conditions, temperatures or high UV exposure may present similar barriers to somatic growth, spread and establishment for M. spicatum (Tucker et al., 2010).

However, Lake Tahoe’s benthic community is currently undergoing significant environmental change (Caires et al., 2013), and eutrophication favors the success of colonists (Christie, Fraser & Nepszy, 1972). Indeed, Lake Tahoe has recently experienced increased disturbance through nearshore development, temperature warming, the establishment of other nonindigenous species (e.g., Asian clam, signal crayfish and various warmwater fishes) and losses in water transparency (Goldman, 1988; Frantz & Cordone, 1996; Chandra et al., 2005; Kamerath, Chandra & Allen, 2008; Coats, 2010; Wittmann et al., 2012). These stressors are likely to alter ecosystem dynamics that may affect the expansion rates of species such as M. spicatum or P. crispus within the lake. The use of relative risk assessments may be a better predictor in the future, when there are fewer barriers to establishment.

Future directions

There are many unknowns associated with the establishment of species, which often leaves managers having to react to, rather than prevent, new infestations of NAS within ecosystems. Here, we have developed an approach to reduce the uncertainty associated with identifying site-specific establishment risk and the subsequent development of surveillance or other management programs within a lake ecosystem. We propose that this framework can also be applied to a wide range of species over multiple spatial scales in part because of the increased availability of species- or system-specific data. Freely available resources that describe species dispersal pathways (e.g., the 100th Meridian Initiative Recreational boater database, National Ballast Information Clearing House) combined with field measurements of physical or biological data (e.g., NOAA National Climatic Data Center, USGS Nonindigenous Aquatic Species Database) can be compiled to build relative risk assessment utilizing the methods similar to those proposed herein.

Supplemental Information

Supplemental Information 1 Recreational boater questionnaire

Questionnaire administered to recreational boaters concerning movements within Lake Tahoe. Question #’s 3, 4, 5 were used in this study.

Click here for additional data file.

Supplemental Information 2 Tahoe Boater Movement Datafile

Travel matrix with named Lake Tahoe nearshore locations. The left column indicates origination location of trip and the top row indicates destination of trip. Cell entries indicate the sum of origination-destinations per location pairing.

Click here for additional data file.

We thank F Davis, J Fram, S MacIntyre, (University of California, Santa Barbara), C Shade (TRPA), CR Goldman, S Hackley, G Schladow, C Strasenburgh and H Segale (University of California, Davis), B Blank and S Chandra (University of Nevada Reno), and N Cartwright and K Boyd (Tahoe Resource Conservation District) for their support of this research. H Mäemets and an anonymous reviewer provided comments that improved the manuscript.

Additional Information and Declarations

Competing Interests

Author Contributions

Lars W.J. Anderson is an employee of WaterweedSolutions (Pt. Reyes, CA).

Marion E. Wittmann conceived and designed the experiments, performed the experiments, analyzed the data, contributed reagents/materials/analysis tools, wrote the paper, prepared figures and/or tables, reviewed drafts of the paper.

Bruce E. Kendall conceived and designed the experiments, analyzed the data, wrote the paper, reviewed drafts of the paper.

Christopher L. Jerde analyzed the data, wrote the paper, reviewed drafts of the paper.

Lars W.J. Anderson analyzed the data, contributed reagents/materials/analysis tools, wrote the paper, reviewed drafts of the paper.

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
