# Peer review of "Estimating relative risk of within-lake aquatic plant invasion using combined measures of recreational boater movement and habitat suitability"

_PeerJ, doi:10.7717/peerj.845_

## Round 0.1 · original submission · Major Revisions

Both reviewers ask for more details on the understanding of dispersion. This needs to be included. Reviewer 2 made many points which should be discussed. I do not think that more experiment is needed, more stats as suggested and more discussion on topics highlighted should be made

·

Basic reporting

Generally meeting the standards; recommendable to add some hydrochemical data into Site description, and about the status of other submergent species.

Experimental design

No comments

Validity of the findings

Conclusions clear, shorter discussion would be better

Comments for the author

General

In the article I suggest to pay attention to the following questions/aspects, not presented yet:
1. Is Myriophyllum spicatum propagating in L Tahoe only by shoots (vegetatively)? According to Preston & Croft (1997) seed germination is very rare.
2. Is L. Tahoe really oligotrophic? Probably it is not very soft-watered lake and so not oligotrophic according all accumulation types (humic compounds, nutrients, carbonates). It is recommendable (very shortly) to show averaged transparency and some general parameters of summer surface layer – alkalinity, TP, TN, Chla as well as depth limit of submerged plants.
3. In site description are named other submergent species but information about their success, possible suppression by M. spicatum, growth areas etc. very scarce. There is a single sentence about dominating of Potamogeton crispus. Probably mechanical stress is influencing other species in the same way? What about the vitality of Myriophyllum quitense? Is the coverage by M. spicatum the largest among submerged species?

Details
• biserial correlation (r. 238) better in Methods?
• ...and r. 286 in Results?
• The sentences in rows 207-209 and 211-212 seem to be a little bit contradictionary
Sentences of the following rows are not understandable:
169-170
214-215
251
265
276-277

It is possible to make a shorter Discussion; about 2/3 of the presented yet.

Reviewer 2 ·

Basic reporting

Review of “Estimating relative risk of within-lake aquatic plant invasion using combined measures of recreational boater movement and habitat suitability” by Wittmann et al.
This work focuses on the stress consecutive to recreational boater movements on the lake Tahoe in USA, induced on the occurrence of populations of the invasive macrophyte Myriophyllum spicatum.
Here only seven years of records are used to establish the predictive model (1995, 1966, 1997, 2000, 2003, 2006 and 2012).
This research matches with hot topics and mainly with the field of managing invasive species in hydrosystems. Assessing the risk of establishment and colonization of an invasive organism is of great interest to the conservation of natural habitats and their sustainability. Vegetative reproduction is certainly the best way for an invasive to colonize new localities as it could occur during the whole year depending on the species like M. spicatum.
Stress generated by boats and their movements can break parts of the macrophytes, initiate the release of propagules and transport these organs away.
My first comment concerns the number of stressful features considered in this study. Indeed, as correlation analyses were performed, it must be of interest to include other physico-chemical parameters and to conduct multivariate correlation analyses to highlight significant relationships between stressful factors that may be independent when considered alone.
Another problem concerns how the species is named into the text. The acronym EWM for Eurasian watermilfoil is incorrect. It is necessary to respect the Latin nomenclature rules and to name the species either Myriophyllum spicatum or M. spicatum.
Poor information is provided concerning the ecological affinities of this taxon, e.g., trophic conditions (N, P, K), pH (mean values), current velocity (mean values), etc. This is necessary for readership unfamiliar with plants and especially macrophytes.
In references, I am not sure that the article “Von Holle B, Simberloff D. 2005. Ecological resistance to biological invasion overwhelmed by propagule pressure. Ecology 86:3212–3218.” could stay in the “H” section. The “V” section should be preferred.

Experimental design

In “materials and methods” section, when authors said that M. spicatum dominates much of the submersed aquatic plant community, particularly in the constructed marinas at the south end of the lake Tahoe, they should indicate the percent of area covered by this species.
Concerning the interview of 10 questions proposed to persons present near the lake in 2005 and 2006, authors should put it in their article (maybe as supplemental material) as it is the only base of boater behavior on this lake. Readership must be aware of the relevancy of the questions.

Validity of the findings

The problem with the wave action is that it was only recorded during 14 days in 2006 (13 localities). Consequently how could these results be relevant?
In the discussion, the main remark concerns the expected ways of propagule dispersal as no correlation is found with recreational boaters. Authors should detailed this point and provide bibliographical references on this subject. Aquatic birds and/or ducks, e.g., mallard, Canada goose, California gull, can transport propagules from other lakes or in other localities of the lake Tahoe.

Comments for the author

My general advice on this article is that authors must strongly complete their study by studying other stress that allow propagules dispersal. Multivariate correlations must be led to assess relationships between colonization of new localities by Myriophyllum spicatum and environmental parameters. Genetic analyses also should be conducted to establish if vegetative reproduction is the only dispersal way here. Finally, studying such macrophytes populations through metapopulation model will allow to understand the behavior of propagules in waters depending on several kinds of parameters, e.g., wind, waves, boaters, animals, altitude, depth, geomorphology, landscape.

---

## Round 0.2 · accepted · Accept

Dear authors. I think you did a great job addressing the reviewer's comments and making your text clearer (and shorter).